# Equality of Opportunity in Supervised Learning

**Moritz Hardt**
Google
m@mrtz.org

**Eric Price**[*]
UT Austin
ecprice@cs.utexas.edu

**Nathan Srebro**
TTI-Chicago
nati@ttic.edu

## Abstract

We propose a criterion for discrimination against a specified sensitive attribute in supervised learning, where the goal is to predict some target based on available features. Assuming data about the predictor, target, and membership in the protected group are available, we show how to optimally *adjust* any learned predictor so as to remove discrimination according to our definition. Our framework also improves incentives by shifting the cost of poor classification from disadvantaged groups to the decision maker, who can respond by improving the classification accuracy.

**We enourage readers to consult the more complete manuscript on the arXiv.**

## 1 Introduction

As machine learning increasingly affects decisions in domains protected by anti-discrimination law, there is much interest in algorithmically measuring and ensuring fairness in machine learning. In domains such as advertising, credit, employment, education, and criminal justice, machine learning could help obtain more accurate predictions, but its effect on existing biases is not well understood. Although reliance on data and quantitative measures can help quantify and eliminate existing biases, some scholars caution that algorithms can also introduce new biases or perpetuate existing ones [1]. In May 2014, the Obama Administration's Big Data Working Group released a report [2] arguing that discrimination can sometimes "be the inadvertent outcome of the way big data technologies are structured and used" and pointed toward "the potential of encoding discrimination in automated decisions". A subsequent White House report [3] calls for "equal opportunity by design" as a guiding principle in domains such as credit scoring.

Despite the demand, a vetted methodology for avoiding discrimination against *protected attributes* in machine learning is lacking. A naïve approach might require that the algorithm should ignore all protected attributes such as race, color, religion, gender, disability, or family status. However, this idea of "fairness through unawareness" is ineffective due to the existence of *redundant encodings*, ways of predicting protected attributes from other features [4].

Another common conception of non-discrimination is *demographic parity* [e.g. 5, 6, 7]. Demographic parity requires that a decision—such as accepting or denying a loan application—be independent of the protected attribute. Through its various equivalent formalizations this idea appears in numerous papers. Unfortunately, the notion is seriously flawed on two counts [8]. First, it doesn't ensure fairness. The notion permits that we accept the qualified applicants in one demographic, but random individuals in another, so long as the percentages of acceptance match. This behavior can arise naturally, when there is little or no training data available for one of the demographics. Second, demographic parity often cripples the utility that we might hope to achieve, especially in the common scenario in which an outcome to be predicated, e.g. whether the loan be will defaulted, is correlated with the protected attribute. Demographic parity would not allow the ideal prediction, namely giving loans exactly to those who won't default. As a result, the loss in utility of introducing demographic parity can be substantial.

---

[*]Work partially performed while at OpenAI.

In this paper, we consider non-discrimination from the perspective of supervised learning, where the goal is to predict a true outcome $Y$ from features $X$ based on labeled training data, while ensuring the prediction is "non-discriminatory" with respect to a specified protected attribute $A$. The main question here, for which we suggest an answer, is what does it mean for such a prediction to be non-discriminatory. As in the usual supervised learning setting, we assume that we have access to labeled training data, in our case indicating also the protected attribute $A$. That is, to samples from the joint distribution of $(X, A, Y)$. This data is used to construct a (possibly randomized) predictor $\hat{Y}(X)$ or $\hat{Y}(X, A)$, and we also use such labeled data to test for non-discriminatory.

The notion we propose is "oblivious", in that it is based only on the joint distribution, or joint statistics, of the true target $Y$, the predictions $\hat{Y}$, and the protected attribute $A$. In particular, it does not evaluate the features in $X$ nor the functional form of the predictor $\hat{Y}(X)$ nor how it was derived. This matches other tests recently proposed and conducted, including demographic parity and different analyses of common risk scores. In many cases, only oblivious analysis is possible as the functional form of the score and underlying training data are not public. The only information about the score is the score itself, which can then be correlated with the target and protected attribute. Furthermore, even if the features or the functional form are available, going beyond oblivious analysis essentially requires subjective interpretation or casual assumptions about specific features, which we aim to avoid.

In a recent concurrent work, Kleinberg, Mullainathan and Raghavan [9] showed that the only way for a meaningful score that is *calibrated within each group* to satisfy a criterion equivalent to equalized odds is for the score to be a perfectly accurate predictor. This result highlights a contrast between equalized odds and other desirable properties of a score, as well the relationship between nondiscrimination and accuracy, which we also discuss.

**Contributions** We propose a simple, interpretable, and easily checkable notion of non-discrimination with respect to a specified protected attributes. We argue that, unlike demographic parity, our notion provides a meaningful measure of discrimination, while allowing for higher utility. Unlike demographic parity, our notion always allows for the perfectly accurate solution of $\hat{Y} = Y$. More broadly, our criterion is easier to achieve the more accurate the predictor $\hat{Y}$ is, aligning fairness with the central goal in supervised learning of building more accurate predictors. Our notion is actionable, in that we give a simple and effective framework for constructing classifiers satisfying our criterion from an arbitrary learned predictor.

Our notion can also be viewed as shifting the burden of uncertainty in classification from the protected class to the decision maker. In doing so, our notion helps to incentivize the collection of better features, that depend more directly on the target rather then the protected attribute, and of data that allows better prediction for all protected classes.

In an updated and expanded paper, `arXiv:1610.02413`, we also capture the inherent limitations of our approach, as well as any other oblivious approach, through a non-identifiability result showing that different dependency structures with possibly different intuitive notions of fairness cannot be separated based on any oblivious notion or test. We strongly encourage readers to consult `arXiv:1610.02413` instead of this shortened presentation.

## 2 Equalized odds and equal opportunity

We now formally introduce our first criterion.

**Definition 2.1** (Equalized odds). We say that a predictor $\hat{Y}$ satisfies *equalized odds* with respect to protected attribute $A$ and outcome $Y$, if $\hat{Y}$ and $A$ are independent conditional on $Y$.

Unlike demographic parity, equalized odds allows $\hat{Y}$ to depend on $A$ but only through the target variable $Y$. This encourages the use of features that relate to $Y$ directly, not through $A$.

As stated, equalized odds applies to targets and protected attributes taking values in any space, including discrete and continuous spaces. But in much of our presentation we focus on binary targets $Y, \hat{Y}$ and protected attributes $A$, in which case equalized odds is equivalent to:

$$\Pr\left\{\hat{Y} = 1 \mid A = 0, Y = y\right\} = \Pr\left\{\hat{Y} = 1 \mid A = 1, Y = y\right\}, \quad y \in \{0, 1\} \qquad (2.1)$$

For the outcome $y = 1$, the constraint requires that $\widehat{Y}$ has equal *true positive rates* across the two demographics $A = 0$ and $A = 1$. For $y = 0$, the constraint equalizes *false positive rates*. Equalized odds thus enforces both equal bias and equal accuracy in all demographics, punishing models that perform well only on the majority.

**Equal opportunity** In the binary case, we often think of the outcome $Y = 1$ as the "advantaged" outcome, such as "not defaulting on a loan", "admission to a college" or "receiving a promotion". A possible relaxation of equalized odds is to require non-discrimination only within the "advantaged" outcome group. That is, to require that people who pay back their loan, have an equal opportunity of getting the loan in the first place (without specifying any requirement for those that will ultimately default). This leads to a relaxation of our notion that we call "equal opportunity".

**Definition 2.2** (Equal opportunity)**.** We say that a binary predictor $\widehat{Y}$ satisfies *equal opportunity* with respect to $A$ and $Y$ if $\Pr\{\widehat{Y} = 1 \mid A = 0, Y = 1\} = \Pr\{\widehat{Y} = 1 \mid A = 1, Y = 1\}$ .

Equal opportunity is a weaker, though still interesting, notion of non-discrimination, and can thus allows for better utility.

**Real-valued scores** Even if the target is binary, a real-valued predictive score $R = f(X, A)$ is often used (e.g. FICO scores for predicting loan default), with the interpretation that higher values of $R$ correspond to greater likelihood of $Y = 1$ and thus a bias toward predicting $\widehat{Y} = 1$. A binary classifier $\widehat{Y}$ can be obtained by thresholding the score, i.e. setting $\widehat{Y} = \mathbb{I}\{R > t\}$ for some threshold $t$. Varying this threshold changes the trade-off between sensitivity and specificity.

Our definition for equalized odds can be applied also to score functions: a score $R$ satisfies equalized odds if $R$ is independent of $A$ given $Y$. If a score obeys equalized odds, then any thresholding $\widehat{Y} = \mathbb{I}\{R > t\}$ of it also obeys equalized odds In Section 3, we will consider scores that might not satisfy equalized odds, and see how equalized odds predictors can be derived from them by using different (possibly randomized) thresholds depending on the value of $A$.

**Oblivious measures** Our notions of non-discrimination are *oblivious* in the following formal sense:

**Definition 2.3.** A property of a predictor $\widehat{Y}$ or score $R$ is said to be *oblivious* if it only depends on the joint distribution of $(Y, A, \widehat{Y})$ or $(Y, A, R)$, respectively.

As a consequence of being oblivious, all the information we need to verify our definitions is contained in the *joint distribution* of predictor, protected group and outcome, $(\widehat{Y}, A, Y)$. In the binary case, when $A$ and $Y$ are reasonably well balanced, the joint distribution of $(\widehat{Y}, A, Y)$ is determined by 8 parameters that can be estimated to very high accuracy from samples. We will therefore ignore the effect of finite sampling and instead assume that we know the joint distribution of $(\widehat{Y}, A, Y)$.

# 3 Achieving non-discrimination

We now explain how to obtain an equalized odds or equal opportunity predictor $\widetilde{Y}$ from a, possibly discriminatory, learned binary predictor $\widehat{Y}$ or score $R$. We envision that $\widehat{Y}$ or $R$ are whatever comes out of the existing training pipeline for the problem at hand. Importantly, we do not require changing the training process, as this might introduce additional complexity, but rather only a post-learning step. Instead, we will construct a non-discriminating predictor which is derived from $\widehat{Y}$ or $R$:

**Definition 3.1** (Derived predictor)**.** A predictor $\widetilde{Y}$ is *derived from a random variable $R$ and the protected attribute $A$* if it is a possibly randomized function of the random variables $(R, A)$ alone. In particular, $\widetilde{Y}$ is independent of $X$ conditional on $(R, A)$.

The definition asks that the value of a derived predictor $\widetilde{Y}$ should only depend on $R$ and the protected attribute, though it may introduce additional randomness. But the formulation of $\widetilde{Y}$ (that is, the function applied to the values of $R$ and $A$), depends on information about the joint distribution of $(R, A, Y)$. In other words, this joint distribution (or an empirical estimate of it) is required at training time in order to construct the predictor $\widetilde{Y}$, but at prediction time we only have access to values of $(R, A)$. No further data about the underlying features $X$, nor their distribution, is required.

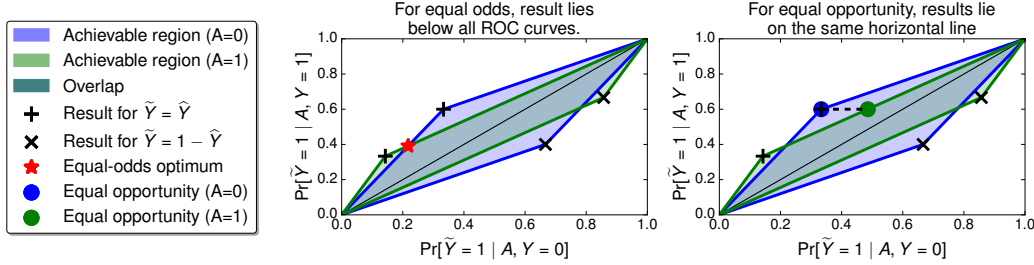

Figure 1: Finding the optimal equalized odds predictor (left), and equal opportunity predictor (right).

**Loss minimization.** It is always easy to construct a trivial predictor satisfying equalized odds, by making decisions independent of $X$, $A$ and $R$. For example, using the constant predictor $\widehat{Y} = 0$ or $\widehat{Y} = 1$. The goal, of course, is to obtain a *good* predictor satisfying the condition. To quantify the notion of "good", we consider a loss function $\ell\colon \{0,1\}^2 \to \mathbb{R}$ that takes a pair of labels and returns a real number $\ell(\widehat{y}, y) \in \mathbb{R}$ which indicates the loss (or cost, or undesirability) of predicting $\widehat{y}$ when the correct label is $y$. Our goal is then to design derived predictors $\widetilde{Y}$ that minimize the expected loss $\mathbb{E}\ell(\widetilde{Y}, Y)$ subject to one of our definitions.

## 3.1 Deriving from a binary predictor

In designing a derived predictor from binary $\widehat{Y}$ and $A$ we can only set four parameters: the conditional probabilities $p_{ya} = \Pr\{\widetilde{Y} = 1 \mid \widehat{Y} = a, A = a\}$. These four parameters, $p = (p_{00}, p_{01}, p_{10}, p_{11})$, together specify the derived predictor $\widetilde{Y}_p$. To check whether $\widetilde{Y}_p$ satisfies equalized odds we need to verify the two equalities specified by (2.1), for both values of $y$. To this end, we denote

$$\gamma_a(\widetilde{Y}) \overset{\text{def}}{=} \left( \Pr\{\widetilde{Y} = 1 \mid A = a, Y = 0\}, \Pr\{\widetilde{Y} = 1 \mid A = a, Y = 1\} \right) . \tag{3.1}$$

The components of $\gamma_a(\widetilde{Y})$ are the *false positive rate* and the *true positive rate* within the demographic $A = a$. Following (2.1), $\widetilde{Y}$ satisfies equalized odds iff $\gamma_0(\widetilde{Y}) = \gamma_1(\widetilde{Y})$. But $\gamma_a(\widetilde{Y}_p)$ is just a linear function of $p$, with coefficients determined by the joint distribution of $(Y, \widehat{Y}, A)$. Since the expected loss $\mathbb{E}\ell(\widetilde{Y}_p, Y)$ is also linear in $p$, we have that the optimal derived predictor can be obtained as a solution to the following linear program with four variables and two equality constraints:

$$\min_{p} \quad \mathbb{E}\ell(\widetilde{Y}_p, Y) \tag{3.2}$$

$$\text{s.t.} \quad \gamma_0(\widetilde{Y}_p) = \gamma_1(\widetilde{Y}_p) \tag{3.3}$$

$$\forall_{y,a} 0 \leqslant p_{ya} \leqslant 1 \tag{3.4}$$

To better understand this linear program, let us understand the range of values $\gamma_a(\widetilde{Y}_p)$ can take:

**Claim 3.2.** $\{\gamma_a(\widetilde{Y}_p) \mid 0 \leqslant p \leqslant 1\} = P_a(\widehat{Y}) \overset{\text{def}}{=} \text{convhull}\left\{ (0,0), \gamma_a(\widehat{Y})\gamma_a(1 - \widehat{Y}), (1,1) \right\}$

These polytopes are visualized in Figure 1. Since each $\gamma_a(\widetilde{Y}_p)$, for each demographic $A = a$, depends on two different coordinates of $p$, the choice of $\gamma_0 \in P_0$ and $\gamma_1 \in P_1$ is independent. Requiring $\gamma_0(\widetilde{Y}_p) = \gamma_1(\widetilde{Y}_p)$ then restricts us exactly to the intersection $P_0 \cap P_1$, and this intersection exactly specifies the range of possible tradeoffs between the false-positive-rate and true-positive-rate for derived predictors $\widetilde{Y}$ (see Figure 1). Solving the linear program (3.2) amounts to finding the tradeoff in $P_0 \cap P_1$ that optimizes the expected loss.

For equalized opportunity, we only require the first components of $\gamma$ agree, removing one of the equality constraints from the linear program. Now, any $\gamma_0 \in P_0$ and $\gamma_1 \in P_1$ that are on the same horizontal line are feasible.

## 3.2 Deriving from a score function

A "protected attribute blind" way of deriving a binary predictor from a score $R$ would be to threshold it, i.e. using $\widehat{Y} = \mathbb{I}\{R > t\}$. If $R$ satisfied equalized odds, then so will such a predictor, and the

optimal threshold should be chosen to balance false and true positive rates so as to minimize the expected loss. When $R$ does not already satisfy equalized odds, we might need to use different thresholds for different values of $A$ (different protected groups), i.e. $\tilde{Y} = \mathbb{I}\{R > t_A\}$. As we will see, even this might not be sufficient, and we might need to introduce randomness also here.

Central to our study is the ROC (Receiver Operator Characteristic) curve of the score, which captures the false positive and true positive (equivalently, false negative) rates at different thresholds. These are curves in a two dimensional plane, where the horizontal axes is the false positive rate of a predictor and the vertical axes is the true positive rate. As discussed in the previous section, equalized odds can be stated as requiring the true positive and false positive rates, $(\Pr\{\widehat{Y} = 1 \mid Y = 0, A = a\}, \Pr\{\widehat{Y} = 1 \mid Y = 1, A = a\})$, agree between different values of $a$ of the protected attribute. That is, that for all values of the protected attribute, the conditional behavior of the predictor is at exactly the same point in this space. We will therefor consider the $A$-conditional ROC curves

$$C_a(t) \stackrel{\text{def}}{=} \left( \Pr\{\widehat{R} > t \mid A = a, Y = 0\}, \Pr\{\widehat{R} > t \mid A = a, Y = 1\} \right).$$

Since the ROC curves exactly specify the conditional distributions $R|A, Y$, a score function obeys equalized odds if and only if the ROC curves for all values of the protected attribute agree, that is $C_a(t) = C_{a'}(t)$ for all values of $a$ and $t$. In this case, any thresholding of $R$ yields an equalized odds predictor (all protected groups are at the same point on the curve, and the same point in false/true-positive plane).

When the ROC curves do not agree, we might choose different thresholds $t_a$ for the different protected groups. This yields different points on each $A$-conditional ROC curve. For the resulting predictor to satisfy equalized odds, these must be at the same point in the false/true-positive plane. This is possible only at points where all $A$-conditional ROC curves intersect. But the ROC curves might not all intersect except at the trivial endpoints, and even if they do, their point of intersection might represent a poor tradeoff between false positive and false negatives.

As with the case of correcting a binary predictor, we can use randomization to fill the span of possible derived predictors and allow for significant intersection in the false/true-positive plane. In particular, for every protected group $a$, consider the convex hull of the image of the conditional ROC curve:

$$D_a \stackrel{\text{def}}{=} \text{convhull}\{C_a(t) \colon t \in [0,1]\} \tag{3.5}$$

The definition of $D_a$ is analogous to the polytope $P_a$ in the previous section, except that here we do not consider points below the main diagonal (line from $(0,0)$ to $(1,1)$), which are worse than "random guessing" and hence never desirable for any reasonable loss function.

**Deriving an optimal equalized odds threshold predictor.** Any point in the convex hull $D_a$ represents the false/true positive rates, conditioned on $A = a$, of a randomized derived predictor based on $R$. In particular, since the space is only two-dimensional, such a predictor $\tilde{Y}$ can always be taken to be a mixture of two threshold predictors (corresponding to the convex hull of two points on the ROC curve). Conditional on $A = a$, the predictor $\tilde{Y}$ behaves as

$$\tilde{Y} = \mathbb{I}\{R > T_a\},$$

where $T_a$ is a randomized threshold assuming the value $\underline{t}_a$ with probability $\underline{p}_a$ and the value $\bar{t}_a$ with probability $\bar{p}_a$. In other words, to construct an equalized odds predictor, we should choose a point in the intersection of these convex hulls, $\gamma = (\gamma[0], \gamma[1]) \in \cap_a D_a$, and then for each protected group realize the true/false-positive rates $\gamma$ with a (possible randomized) predictor $\tilde{Y}|(A = a) = \mathbb{I}\{R > T_a\}$ resulting in the predictor $\tilde{Y} = \Pr \mathbb{I}\{R > T_A\}$. For each group $a$, we either use a fixed threshold $T_a = t_a$ or a mixture of two thresholds $\underline{t}_a < \bar{t}_a$. In the latter case, if $A = a$ and $R < \underline{t}_a$ we always set $\tilde{Y} = 0$, if $R > \bar{t}_a$ we always set $\tilde{Y} = 1$, but if $\underline{t}_a < R < \bar{t}_a$, we flip a coin and set $\tilde{Y} = 1$ with probability $\underline{p}_a$.

The feasible set of false/true positive rates of possible equalized odds predictors is thus the intersection of the areas under the $A$-conditional ROC curves, and above the main diagonal (see Figure 2). Since for any loss function the optimal false/true-positive rate will always be on the upper-left boundary of this feasible set, this is effectively the ROC curve of the equalized odds predictors. This ROC curve is the pointwise minimum of all $A$-conditional ROC curves. The performance of

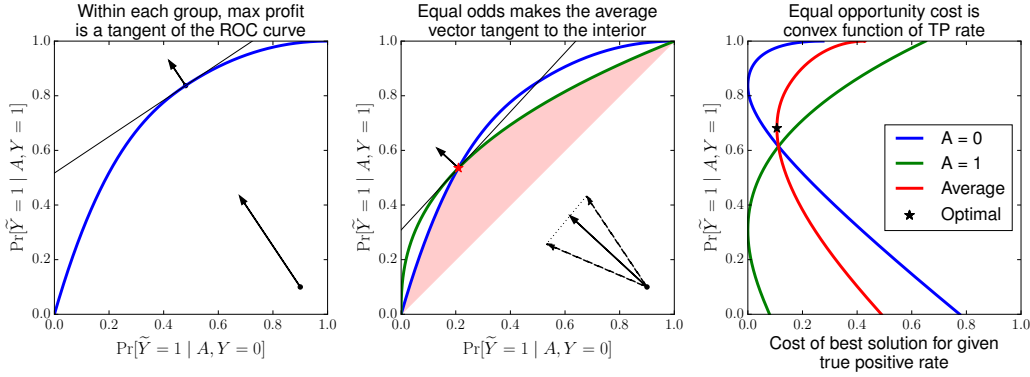

Figure 2: Finding the optimal equalized odds threshold predictor (middle), and equal opportunity threshold predictor (right). For the equal opportunity predictor, within each group the cost for a given true positive rate is proportional to the horizontal gap between the ROC curve and the profit-maximizing tangent line (i.e., the two curves on the left plot), so it is a convex function of the true positive rate (right). This lets us optimize it efficiently with ternary search.

an equalized odds predictor is thus determined by the minimum performance among all protected groups. Said differently, requiring equalized odds incentivizes the learner to build good predictors for *all* classes. For a given loss function, finding the optimal tradeoff amounts to optimizing (assuming w.l.o.g. $\ell(0,0) = \ell(1,1) = 0$):

$$\min_{\forall a \,:\, \gamma \in D_a} \gamma[0]\ell(1,0) + (1 - \gamma[1])\ell(0,1) \tag{3.6}$$

This is no longer a linear program, since $D_a$ are not polytopes, or at least are not specified as such. Nevertheless, (3.6) can be efficiently optimized numerically using ternary search.

**For an optimal equation opportunity derived predictor** the construction is similar, except its sufficient to find points in $D_a$ that are on the same horizontal line. Assuming continuity of the conditional ROC curves, we can always take points on the ROC curve $C_a$ itself and no randomization is necessary.

## 4 Bayes optimal predictors

In this section, we develop the theory a theory for non-discriminating Bayes optimal classification. We will first show that a Bayes optimal equalized odds predictor can be obtained as an derived threshold predictor of the Bayes optimal regressor. Second, we quantify the loss of deriving an equalized odds predictor based on a regressor that deviates from the Bayes optimal regressor. This can be used to justify the approach of first training classifiers without any fairness constraint, and then deriving an equalized odds predictor in a second step.

**Definition 4.1** (Bayes optimal regressor). Given random variables $(X, A)$ and a target variable $Y$, the *Bayes optimal regressor* is $R = \arg\min_{r(x,a)} \mathbb{E}\left[(Y - r(X, A))^2\right] = r^*(X, A)$ with $r^*(x, a) = \mathbb{E}[Y \mid X = x, A = a]$.

The Bayes optimal classifier, for any proper loss, is then a threshold predictor of $R$, where the threshold depends on the loss function (see, e.g., [10]). We will extend this result to the case where we additionally ask the classifier to satisfy an oblivious property:

**Proposition 4.2.** *For any source distribution over $(Y, X, A)$ with Bayes optimal regressor $R(X, A)$, any loss function, and any oblivious property $C$, there exists a predictor $Y^*(R, A)$ such that:*

1. *$Y^*$ is an optimal predictor satisfying $C$. That is, $\mathbb{E}\ell(Y^*, Y) \leqslant \mathbb{E}\ell(\widehat{Y}, Y)$ for any predictor $\widehat{Y}(X, A)$ which satisfies $C$.*

2. *$Y^*$ is derived from $(R, A)$.*

**Corollary 4.3** (Optimality characterization). *An optimal equalized odds predictor can be derived from the Bayes optimal regressor $R$ and the protected attribute $A$. The same is true for an optimal equal opportunity predictor.*

We can furthermore show that if we can approximate the (unconstrained) Bayes optimal regressor well enough, then we can also construct a nearly optimal non-discriminating classifier:

**Definition 4.4.** We define the *conditional Kolmogorov distance* between two random variables $R, R' \in [0, 1]$ in the same probability space as $A$ and $Y$ as:

$$d_{\mathrm{K}}(R, R') \stackrel{\mathrm{def}}{=} \max_{a,y\in\{0,1\}} \sup_{t\in[0,1]} |\Pr\{R > t \mid A = a, Y = y\} - \Pr\{R' > t \mid A = a, Y = y\}| . \tag{4.1}$$

**Theorem 4.5** (Near optimality). *Assume that $\ell$ is a bounded loss function, and let $\widehat{R} \in [0, 1]$ be an arbitrary random variable. Then, there is an optimal equalized odds predictor $Y^*$ and an equalized odds predictor $\widehat{Y}$ derived from $(\widehat{R}, A)$ such that*

$$\mathbb{E}\ell(\widehat{Y}, Y) \leqslant \mathbb{E}\ell(Y^*, Y) + 2\sqrt{2} \cdot d_{\mathrm{K}}(\widehat{R}, R^*) ,$$

*where $R^*$ is the Bayes optimal regressor. The same claim is true for equal opportunity.*

# 5   Case study: FICO scores

FICO scores are a proprietary classifier widely used in the United States to predict credit worthiness [11]. These scores, ranging from 300 to 850, try to predict credit risk; they form our score $R$. People were labeled as in *default* if they failed to pay a debt for at least 90 days on at least one account in the ensuing 18-24 month period; this gives an outcome $Y$. Our protected attribute $A$ is race, which is restricted to four values: Asian, white non-Hispanic (labeled "white" in figures), Hispanic, and black. FICO scores are complicated proprietary classifiers based on features, like number of bank accounts kept, that could interact with culture and race.

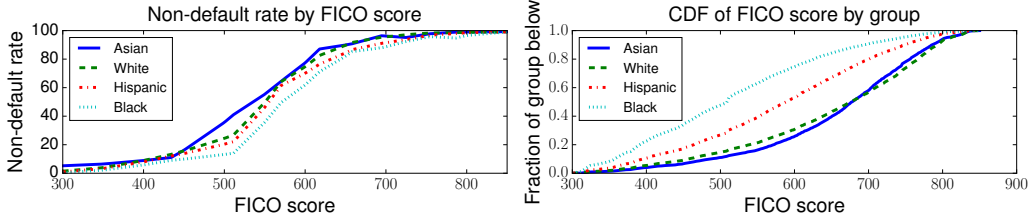

Figure 3: These two marginals, and the number of people per group, constitute our input data.

To illustrate the effect of non-discrimination on utility we used a loss in which false positives (giving loans to people that default on any account) is 82/18 as expensive as false negatives (not giving a loan to people that don't default). Given the marginal distributions for each group (Figure 3), we can then study the optimal profit-maximizing classifier under five different constraints on allowed predictors:

- **Max profit** has no fairness constraints, and will pick for each group the threshold that maximizes profit. This is the score at which 82% of people in that group do not default.
- **Race blind** requires the threshold to be the same for each group. Hence it will pick the single threshold at which 82% of people do not default overall.
- **Demographic parity** picks for each group a threshold such that the fraction of group members that qualify for loans is the same.
- **Equal opportunity** picks for each group a threshold such that the fraction of *non-defaulting* group members that qualify for loans is the same.
- **Equalized odds** requires both the fraction of non-defaulters that qualify for loans and the fraction of defaulters that qualify for loans to be constant across groups. This might require randomizing between two thresholds for each group.

Our proposed fairness definitions give thresholds between those of max-profit/race-blind thresholds and of demographic parity. Figure 4 shows the thresholds used by each predictor, and Figure 5 plots the ROC curves for each group, and the per-group false and true positive rates for each resulting predictor. Differences in the ROC curve indicate differences in predictive accuracy between groups (*not* differences in default rates), demonstrating that the majority (white) group is classified more accurately than other.

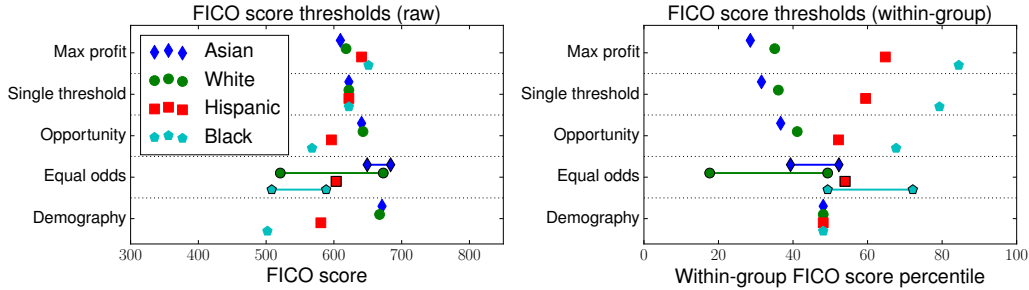

Figure 4: FICO thresholds for various definitions of fairness. The equal odds method does not give a single threshold, but instead $\Pr[\widehat{Y} = 1 \mid R, A]$ increases over some not uniquely defined range; we pick the one containing the fewest people.

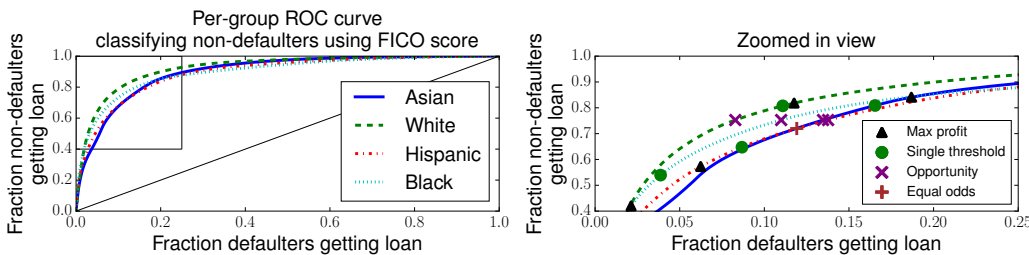

Figure 5: The ROC curve for using FICO score to identify non-defaulters. Within a group, we can achieve any convex combination of these outcomes. Equality of opportunity picks points along the same horizontal line. Equal odds picks a point below all lines.

We can compute the profit achieved by each method, as a fraction of the max profit achievable. A race blind threshold gets 99.3% of the maximal profit, equal opportunity gets 92.8%, equalized odds gets 80.2%, and demographic parity only 69.8%.

# 6   Conclusions

We proposed a fairness measure that accomplishes two important desiderata. First, it remedies the main conceptual shortcomings of demographic parity as a fairness notion. Second, it is fully aligned with the central goal of supervised machine learning, that is, to build higher accuracy classifiers.

Our notion requires access to observed outcomes such as default rates in the loan setting. This is precisely the same requirement that supervised learning generally has. The broad success of supervised learning demonstrates that this requirement is met in many important applications. That said, having access to reliable "labeled data" is not always possible. Moreover, the measurement of the target variable might in itself be unreliable or biased. Domain-specific scrutiny is required in defining and collecting a reliable target variable.

Requiring equalized odds creates an incentive structure for the entity building the predictor that aligns well with achieving fairness. Achieving better prediction with equalized odds requires collecting features that more directly capture the target, unrelated to its correlation with the protected attribute. An equalized odds predictor derived from a score depends on the pointwise minimum ROC curve among different protected groups, encouraging constructing of predictors that are accurate in all groups, e.g., by collecting data appropriately or basing prediction on features predictive in all groups.

An important feature of our notion is that it can be achieved via a simple and efficient post-processing step. In fact, this step requires only aggregate information about the data and therefore could even be carried out in a privacy-preserving manner (formally, via Differential Privacy).

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
