[Reviews · NeurIPS 2016]

Reviewer 1

Summary

This paper treats an incredibly important and foundational problem (fairness), proposes a creative but simple new definition, gives techniques for achieving the definition, proves theorems with regards to optimality, and even provides empirical results.

Qualitative Assessment

This paper is excellent. It treats an incredibly important and foundational problem (fairness), proposes a creative but simple new definition, gives techniques for achieving the definition, proves theorems with regards to optimality, and even provides empirical results. As learning algorithms are used more and more broadly in situations where their decisions affect people’s lives, fairness of these algorithms becomes a critical technical, social, and legal problem. While there is certainly no single “right” definition and paradigm when it comes to fairness, this definition seems to clearly be *a* right definition. It’s so clean and simple that in retrospect, it seems obvious—a sign of an excellent idea. One of the many things I love about this definition and this work is how it shifts the structure of power and incentives—once a learner is constrained to be fair, under either of the definitions proposed, she is immediately incentivised to gather more data or make other efforts to do a better job of understanding protected populations. I expect that the ideas from this paper will be highly influential, and this work raises many interesting issues (which I regard as yet another strength): how to regulate fairness, detect unfairness, how firms might attempt to evade regulation, social and legal consequences, how one ought to select among the polytope of fair outcomes (secondary concerns beyond profit), … Some additional comments: I would have preferred that the proof of Lemma 3.3 be fleshed out more in the appendix, making the definition of a derived predictor explicit in terms of a point in each of the two polytopes. For an outsider, it was a bit non-intuitive to reconstruct the proof. I would like the appendix to contain more discussion of the figures. The discussion was far too terse and difficult to unpack, and these are very useful figures. In Section 3.2, there seem to be implicit assumptions that \hat{R} is monotone in Y (in order to get the statement about threshold-based predictors) and that \hat{R} is right more than it is wrong (in order to justify cutting off the region below the line from (0,0) to (1,1) from the polytope. p. 4 line 141 shouldn’t “false negative” be “true positive”? p. 4 line 159 statistic statistic

Confidence in this Review

3-Expert (read the paper in detail, know the area, quite certain of my opinion)


Reviewer 2

Summary

Fairness in prediction is a problem that is becoming more and more acute nowadays. Suppose we are looking to find some classifier Z which we want to be fair towards a "protected" attribute A (e.g., race). The naive approach is to require that given Pr[Z=1 | A=a] is the same for all values that A take, so that from knowing the classification of a datapoint leads to no advantage in guessing the point's attribute. This paper introduces an alternative notion of fairness. Given a sensitive attribute an a true label Y, then the classifier "equalizes odds" if for any value of y we have that Pr[Z=1 | Y=y, A=a] = Pr[Z=1 | Y=y, A=a'] for any two values a, a' of A. (It "equalizes opportunity" if the above holds solely for Y=1.) Hence, knowing the prediction gives no more advantage in inferring the sensitive attribute when one also knows the outcome Y. The paper then goes to proving that post-processing a Z into a Z' that satisfies either property is feasible in poly-time, given the contingency table of (Z, Y, A), for binary predictors and also for predictors Z that use a "score" in the range [0,1] to classify. They also bound the "price" of fairness in the base of Bayesian classification and finally they test their approach on a FICO data of credit scores and the loaners who actually defaulted on their loans, in partition to race. None-surprisingly, the equal-odds classifier does slightly poorly than the non-fair classifier, but clearly outperforms the classifiers that preserve the naive notion of fairness.

Qualitative Assessment

This is a highly novel approach for fairness that is likely to have a great impact on machine learning. The paper is extremely interesting, very novel, has some nice theoretical results and interesting experiment, and moreover - it leads to *MANY* follow-up questions. I thus recommend it for full oral presentation. My only complaint regarding the paper is regarding its style and presentation. I feel that a more detailed version (in the supplementary material section) with clarifications and *better figures* would have resulted in a much clearer paper. I had hard time following the proofs (which aren't wrong, just require some non-trivial deductions and keeping track of notation), and clearly, a longer pass at the paper would have contributed greatly to a better understanding. I urge the authors to do so, as I think this paper is clearly "one for the books" and I suspect it will be studied in depth in the future. A few more specific styling comments: * Equation 3.1: "Similarly the second component is the false negative rate" --- isn't it the true positive rate? Isn't false negative rate be the second coordinate of \gamma_a(1-\hat Y)? * Lemma 3.2 (nitpicking) use parentheses for second coordinate $(\gamma_i(\hat Y))_2$ * Lemma 3.2, below: a bit more discussion on this polytope wouldn't hurt. In particular -- why (0,0) & (1,1) are there? (They are the "demographic classifiers", but why are they required? I couldn't see it from the proof of 3.3 in A.1) If \hat Y is multi-labeled, would you put all \gamma_a(Y=y) points? * Figure 1, legend: aren't the equal opportunity points w.r.t to \tilde Y? (i.e., it's \tilde Y=0 and \tilde Y = 1?) If not, then I really have missed something at your explanation and/or in the understanding of the figure... * Figure 2 completely confused me. Its explanation is baffling, and it seems unrelated to any of the text that actually appears in Section 3.2. Consider replacing it or putting it in the appendix with *MUCH* more instructions as to how the reader should interpret it. * The curve C_a(t) --- seems like, from the figure, it is (Pr[R < t | ...], Pr[R < t | ...]), or I just confused because you drew in Section 5 the CDFs? * Equation (3.3) unclear what you mean by convex-hull - especially since Figure 2 has the entire area below the curve highlighted, but then you say in line 176 that you ignore points below the main diagonal. (Am I right in understanding they are "bad than a random guess" because such points give pts with Y=0 better probability of getting classified as positive than pts with Y=1? This too, which seems simple, required me to parse through some definitions - which would have been far easier if you left room for more discussion...) * Figure 3 contributes nothing to the discussion. I suggest removing it. * Proof of 4.3: why not run the conversion proving (2) for every Y\in C without changing the classification cost, hence the conversion also holds for the best Y? * Figure 6: hard to parse. What is "Raw" and what is "within group"? * Proof of Proposition 3.4: Why not simply write the LP itself? Moreover, the proof gives the impression you just need variables for Pr[\tilde Y=\hat Y], but without dependence on A you cannot write the constraint \gamma_0(\tilde Y)=\gamma_1(\tilde Y). * Proof of Thm 4.5, line 395 --- it is not easy for me. I had a hard time following... This is an example where pic would be useful...

Confidence in this Review

2-Confident (read it all; understood it all reasonably well)


Reviewer 3

Summary

This paper studies the problem of constructing fair machine learning model in the sense that these model should avoid biases such as race or gender. The paper proposes a definition of fairness and studies theoretical implementations of the definition.

Qualitative Assessment

I think the abstract should be re-written: the current version does not explain what is the problem the paper is trying to solve. Since the concept of “fairness” in learning is not known to the typical NIPS reader some intuition about it should be provided here. The problem of fairness is of great importance. For example, consider the scenario of candidates for a job, you would like to have such a decision done in a way that is unbiased by gender and race. However, since humans are influenced by these biases, if you were to train a machine learning model to make hiring decisions, it will have the same bias. Unfortunately, this work does not seem to solve this problem since they assume that the training data is unbiased (see line 51). I find the paper hard to follow. Although the authors try to provide intuition to the definitions, I failed to get to the heart of it. For example, they define the term D_a (eq (3.3). In the verbal description they restrict it to be above the diagonal but in the formal definition they do not. Therefore, it is not clear which version is used in eq (3.4). Furthermore, in (3.4) it is not clear why is it necessary to restrict the solution using D_a. More broadly, I find the notation used confusing, especially the fact that the dependency of Y-tilde in Y-hat, A and other factors is omitted. I had to constantly remind myself that Y-tilde is a function. 1. Line 51: you assume equal true positive rate, why is this assumption valid? Isn’t it essentially doing demographic parity? 2. Line 141: isn’t the second term the true rate? 3. Section 3.2: in section 3.1 you showed a generic method that is independent of the data representation while in section 3.2 you look at the specific case of threshold functions. Why wouldn’t you use the method of 3.1? 4. I find (3.4) to be confusing. Y-tilde is a function of R, A, Y and some random bits. 5. In (3.4), what would happen if you remove the constraint that gamma_a is in D_a? I think that the graph for Y-tilde cannot be above the ROC curve and would not be below the diagonal because of the optimization of the loss. If I got it right, it means that Y-tilde should be such that the ROC curve, conditioned on class A=0 should be the same as the ROC curve conditioned on A=1. After reading the comments of the authors, I still thing that the assumption that Y is unbiased is a major flaw. However, I think that this paper will inspire discussions and further research. Hence, as a conference paper, it is a good contribution.

Confidence in this Review

1-Less confident (might not have understood significant parts)


Reviewer 4

Summary

The paper proposes a new measure for fairness in classification, provide some theoretical results on how this might be achieved with a classifier, and present a case study using FICO scores. The work sits in the space of "designing fair classifiers" with a new notion of 'fair'.

Qualitative Assessment

This is a solid contribution to the realm of fairness-aware learning. While the idea of defining fairness based on equal error treatment of groups is not new (It's buried in the Barocas/Selbst work and is implied by the recent ProPublica article on the COMPAS system for sentencing - something the authors should cite in future versions of this paper), it hasn't been worked out as an explicit measure before, and this paper is a good step towards it. I'd quibble slightly about the fragility of the idea of requiring equality of the odds ratio. I think a more general approach might look at other measures over the 2x2 table and also take into account more systematically the non-binary nature of the domains. I'm also a little confused about the main results: while the authors prove the existence of Bayes optimal classifiers that achieve the desired fairness goals, I had a hard time figuring out exactly where the algorithms were. The experiments had some interesting issues. In particular, I was looking at Figure 8, and the different behavior of the different methods for different ethnic groups. For example demographic parity and the proposed measures have different behavior for different groups. I'd have liked to see some discussion of this phenomenon, which could possible lead to a better discussion of the pros and cons of the different measures. I'd also suggest that the authors don't need to position their approach in opposition to measures like demographic parity, given that there are many situations where that measure is indeed appropriate, (I grant that there are places where it has problems). It worries me somewhat that the authors don't consider the vast array of other approaches to designing fair classifiers and evaluate how well they do. Their case study is narrowly limited to a threshold classifier, and it's not clear what would happen when you have more complex classiifiers: for example, the authors haven't considered the recent work of Zafar et al (https://www.mpi-sws.org/~gummadi/papers/fatml_15.pdf). They should also look into the paper by Kun et al (http://arxiv.org/abs/1601.05764). Ultimately I'm a little torn. I think the ideas here are quite nice and have a lot of promise. But there's important due diligence that needs to be done.

Confidence in this Review

3-Expert (read the paper in detail, know the area, quite certain of my opinion)


Reviewer 5

Summary

The paper explores several standards of fairness for classification tasks (max product, racial blindness, equality of opportunity, equality of odds, demographic parity), frames equalized odds as an optimization problem, gives theoretical guarantees for a variety of categories of classifier, and concludes with a case study based on FICO scores.

Qualitative Assessment

The paper makes a novel contribution and makes it well. A conclusion or future work section would have been appreciated, but it appears as though space was a constraint. There's some question in my mind as to whether an equalized odds solution that could only perform as accurately as the hardest group to classify would be considered fair. To follow the authors' argument, if we had an oracle only for some groups, we could not use that oracle. Also, if we believe there are fundamental inequalities within the groups in question, it's odd to think we'd equalize odds by increasing the false positives rate within one group (and thereby increase the resources given to that group in the name of fairness). This suggests another regime where we only have inequality constraints on the false positives rate.

Confidence in this Review

2-Confident (read it all; understood it all reasonably well)


Reviewer 6

Summary

This paper tracked an important problem of ensuring fairness in supervised learning in which the predictions are not dependent on the protected attributes conditioned on the true target. The authors proposed a new definition of fairness, and a framework that post-processes the binary predictor, or binary threshold predictor, or Bayes optimal predictor so that their fairness conditions are satisfied. They showed that these post-processing are optimal or near optimal in terms of minimizing the loss function under the fairness constraints. The experimental results showed that the proposed methods outperform demographic parity while ensuring the defined fairness constraints.

Qualitative Assessment

In lines 31-32, you say "the notion permits that we accept qualified applicants in the demographic A = 0, but unqualified individuals in A = 1, so long as the percentages of acceptance match". But I can't understand that the situation does not ensure the fairness. The concept of the equalized odds is not clear for me. Why can the prediction \hat{Y} depend on A through the target variable Y? This definition may implicitly make an assumption that the true target variable Y is fair. But I think the training dataset can consist of unfair target, and thus the true target variable Y is implicitly unfair. Therefore, the predictor that satisfies the equal odds can make unfair treatment. You should more explain the reason why can the prediction depend on the protected attribute through the true target. I am not convinced with the definition of the equal opportunity. This definition requires an fair opportunity for people who pay back their loan, but not for people who does not pay back their loan. It allows that people with A=0 have move opportunity to get the loan than people with A=1 if they will not pay back their loan. While the fairness definitions have some flaws, the proposed methods are very interesting. Especially, the remark on the lower bound in line 218 is important for more realistic situation that the contingency tables are constructed from the finite samples. It is nice to show more precise claim for that.

Confidence in this Review

2-Confident (read it all; understood it all reasonably well)